# Separable Confident Transductive Learning for Dairy Cows Teat-End Condition Classification

**DOI:** 10.3390/ani12070886

**Published:** 2022-03-31

**Authors:** Youshan Zhang, Ian R. Porter, Matthias Wieland, Parminder S. Basran

**Affiliations:** 1Department of Clinical Sciences, College of Veterinary Medicine, Cornell University, Ithaca, NY 14853, USA; irp5@cornell.edu (I.R.P.); psb92@cornell.edu (P.S.B.); 2Department of Population Medicine and Diagnostic Sciences, College of Veterinary Medicine, Cornell University, Ithaca, NY 14853, USA; mjw248@cornell.edu

**Keywords:** transductive learning, dairy cows, teat-end health assessments

## Abstract

**Simple Summary:**

The health of dairy cows is important for milk quality and the health of the mammary gland. Traditionally, teat-end health has been assessed manually through visual inspection of teat-end callosity thickness and roughness (i.e., hyperkeratosis), which is a risk-factor for mastitis. Here, we describe a computer-vision approach to replace the time-consuming and expensive manual assessment of teat-end hyperkeratosis. Using separable confident transductive learning, a convolutional neural network is trained with the goal of increasing the feature differences in the images of teat-ends with different classifications of hyperkeratosis. When compared with the traditional approach of transfer learning of a convolution neural network for classifying the extent of hyperkeratosis, the overall accuracy of our model increased from 61.8 to 77.6%. This substantial improvement in accuracy renders the possibility of using image-based machine learning to routinely monitor hyperkeratosis on commercial dairy farm settings.

**Abstract:**

Teat-end health assessments are crucial to maintain milk quality and dairy cow health. One approach to automate teat-end health assessments is by using a convolutional neural network to classify the magnitude of teat-end alterations based on digital images. This approach has been demonstrated as feasible with GoogLeNet but there remains a number of challenges, such as low performance and comparing performance with different ImageNet models. In this paper, we present a separable confident transductive learning (SCTL) model to improve the performance of teat-end image classification. First, we propose a separation loss to ameliorate the inter-class dispersion. Second, we generate high confident pseudo labels to optimize the network. We further employ transductive learning to narrow the gap between training and test datasets with categorical maximum mean discrepancy loss. Experimental results demonstrate that the proposed SCTL model consistently achieves higher accuracy across all seventeen different ImageNet models when compared with retraining of original approaches.

## 1. Introduction

Mastitis remains one of the most frequently occurring diseases in dairy cows, often arising from intramammary infections by way of the teat canal. Machine milking can affect teat canal integrity and lead to increased teat-end callosity, which can increase the risk of bacterial infections of the mammary gland  [1]. Frequent monitoring of teat-end callosity is critical for a mastitis prevention program [2]. However, cow-side manual assessments of teat-health, which is the current best practice, is time-consuming and suffers from inter- and intra-rater variability [3]. Another challenge is the inability to assess the entire herd in large dairy farms. To address some of these challenges, deep learning (DL) has been proposed where GoogLeNet transfer learning was used to classify the extent of hyperkeratosis using a four-level classification scheme [4]. The overall accuracy of this approach was 46.7–61.8%, suggesting feasibility but, as of yet, insufficient accuracy to be useful as a clinical decision tool. As shown in Figure 1 with a t-SNE map, the training (red) and test (blue) data are observed as mixed together after retraining GoogLeNet and unable to discriminate the four classes. The indistinct boundaries of the classes lead to another challenge to improve the performance of the teat-end condition classification problem.

In this paper, we propose a new paradigm that yields a substantial improvement in accuracy of teat-end image classification while retaining the flexibility and accessibility of commonly used ImageNet classifiers such as AlexNet [5], GoogLeNet [6], Xecption [7], and NasNetLarge [8]. To address the aforementioned challenges, we aggregate four different loss functions in one framework: classification loss, separation loss, pseudo labeled test data classification loss, and categorical maximum mean discrepancy (MMD) loss. As shown in Figure 2, using these proposed novel loss functions, our model can realize the inter-class dispersion and intra-class compactness. This paper provides three specific contributions:We propose a novel separable confident transductive learning model (SCTL) to improve accuracy for the teat-end image classification. To improve the discrimination of different classes, we first propose a separation loss to enlarge the dissimilarity between different categories.We develop a pseudo labeling adjustment learning paradigm to continuously generate high confidence examples for the test data and further optimize the network with test data information.We narrow the gap between intra-class differences between training and test data with transductive learning by minimizing categorical MMD loss and further align the condition distribution between training and test data.

In this study, we performed experiments with seventeen benchmark ImageNet models by optimizing these loss functions and increasing the way that differences are detected between the images. The accuracy of our SCTL model with GoogLeNet is increased from 61.8 to 77.6%. This substantial increase in accuracy may render image-based hyperkeratosis classifications feasible on commercial dairy farm settings.

## 2. Related Work

### 2.1. Teat-End Classification

In the dairy industry, mastitis, which is an inflammation of one or more of the cow’s mammary glands, is a frequently occurring disease that affects dairy cow health and milk quality. Mechanical stresses on the cow’s teat-end can evoke circulatory changes to it and, over the course of several weeks, can result in increased teat-end callosity thickness and roughness [10]. These changes to the teat-end can increase the risk of pathogenic bacteria infiltrating the cow’s udders. To monitor and reduce these risks, regular inspections of the dairy cow’s teat-end health is recommended [11]. This is achieved by manually inspecting the teat-ends of at least 20% of the cows in the herd [12]; however, herd-level assessments are time-consuming, expensive, imprecise, and subjective [3]. Four classes scoring of hyperkeratosis is a standard classification that is usually used in cow teat-end classification (Score 1: no ring; Score 2: smooth ring; Score 3: rough ring; Score 4: very rough ring) [13].

Transfer learning has been applied in various computer vision tasks whose performance relies on the diversity of image data and fine-tuning of network parameters. With the invention of different ImageNet models, transfer learning has been widely adapted in image classification, objection detection, and segmentation problems. Porter et al. [4] recently described a machine-learning approach where images of teat-ends could be used to train a convolution neural network (CNN), such as GoogleNet, to classify the extent of teat-end hyperkeratosis using a four-point scoring system. Although the overall accuracy of our original approach on test data showed promise of automatic teat-end assessment, the accuracy was relatively low, which highlighted the need for improvement.

### 2.2. Transductive Learning

Transductive learning (TL) is a process that trains both labeled training data and unlabeled test data [14,15]. It is generally used in semi-supervised learning scenarios. Different from frequently used supervised inductive classification, which aims to train a classification model based on the labeled training data to approximate test data class distribution, the goal of transductive learning is to find an admissible function using the unlabeled data to improve classification performance [16]. The key idea of TL is that the predicted labels for the test samples are viewed as optimization variables, which can be iteratively updated in the training process [17]. When TL is applied, it is often assumed that the training and test sets share a similar distribution [18]. As shown in Figure 1, the training and test sets are not separated, suggesting they are sampled from the same distribution. This observation has motivated us to employ TL for teat-end image classification and, as of yet, has not yet been explored in our specific problem.

### 2.3. Pseudo Labeling

The purpose of pseudo labeling is to seek the generation of labels or pseudo labels for unlabeled data to guide the learning process [19]. Pseudo labeling typically generates pseudo labels for the unlabeled data either based on hard assigned labels (the predictions from neural network [17,20]) or the predicted class probability [21,22,23]. Under such a regime, label information from unlabeled data can be included during training. In deep networks, the classifier from the training data is usually treated as an initial pseudo labeler to generate the pseudo labels for the test data (and use them as if they were real labels). There are several algorithms for obtaining pseudo labels and promote the performance of unlabeled data. Xie et al. [22] proposed a Moving Semantic Transfer Network (MSTN) to develop semantic matching and domain adversary losses to obtain pseudo labels. Iscen et al. [24] assigned pseudo labels to unlabeled samples based on neighborhood graphs. Zhang et al. [25] offer a label propagation with augmented anchors method to improve label propagation via the generation of unlabeled virtual samples with label prediction. Haase et al. [26] trained re-initialized networks and unlabeled datasets on each partition. The trained networks were used to filter the labels for training the newer networks. However, most of their experiments are conducted based on noisy data. Although previous pseudo labeling approaches are general and domain-agnostic, they tend to underperform since noisy pseudo labeled samples degrade model performance. In addition, most pseudo labeling methods employ a two-stage paradigm. The pseudo labels in the first stage (using the trained training data classifier) are generated and then used to train the model along with the labeled training data in the second stage. Our work differs from these approaches by generating high confidence examples with adjustment learning using a novel scheme, which allows for competitive results for teat-end image classification.

## 3. Methods

### 3.1. Motivation

The scientific goal of our paper is to develop a fully automated deep learning model that can accurately identify different categories of dairy cow teat-end conditions. The utilitarian goal is to detect hyperkeratosis of the teat-end area (Score 3 and Score 4) in the commercial dairy farm setting. Our research problem is the teat-end image classification task, and we aim to improve the classification accuracy using transductive learning and pseudo labeling.

### 3.2. Problem

Let D be a dataset and subscripts R and T refer to training or testing subsets of the data. Image classification can be formulated as the problem of learning a classifier *f* from a set of training data, $D_{R}={\left\{\left(X_{R}^{i},Y_{R}^{i}\right)\right\}}_{i=1}^{N_{R}}$, where $y_{i}$ is the ground-truth label in *C* categories corresponding to $x_{i}$, and $N_{R}$ is the number of samples in the training dataset. In our setting, *f* is a classifier from the CNN. The goal of a vanilla image classification problem is to improve the accuracy of the unlabeled T dataset examples: $D_{T}={\left\{\left(X_{T}^{j},Y_{T}^{j}\right)\right\}}_{j=1}^{N_{T}}$. However, due to the diversity of the datasets and fuzzy differences between different categories, the accuracy of test samples remains difficult to improve.

### 3.3. Transfer Learning

With the emergence of different ImageNet models, fine-tuning one of the ImageNet models with transfer learning is often applied in classifying new datasets. The parameters of these different ImageNet models are fit by optimizing a typical categorical cross-entropy (CE) loss function L
(1)$$L_{CE}=-\frac{1}{N_{R}}\sum_{i=1}^{N_{R}}\sum_{c=1}^{C}Y_{R_{c}}^{i}\log\left(f_{c}\left(X_{R}^{i}\right)\right),$$
where $Y_{R_{c}}^{i}\in {\left[0,1\right]}^{C}$ is the binary indicator of each class *c* in true label for observation $\Phi (X_{S}^{i})$, and $f_{c}\left(X_{R}^{i}\right)$ is the predicted probability of class *c* using classifier *f*.

### 3.4. Separation Loss

As shown in Figure 1, different classes of training and test datasets are mixed together. The decision boundary for the trained network remains fuzzy, leading to poor model performance and low accuracy of teat-end classification. Hence, it is necessary to improve the discrimination between different classes.

The purpose of a new separation loss function, $L_{S}$, is to improve the inter-class dispersion so that the boundaries between different categories can be separable, and samples in the same categories can be more closely associated with each other. The core part of separation loss is to reduce the similarity between different classes. Since the network is trained using batch-wise samples, we inevitably encounter situations where the number of samples in different classes are imbalanced. We hence calculate the covariance matrix of the output of each categories’ samples and then minimize the structural similarity [27] between each two categories’ covariance matrix as follows.
(2)$$\begin{array}{cc} \; & L_{S}=\frac{1}{\sum_{i=1}^{C-1}\sum_{j=i+1}^{C}}\sum_{i=1}^{C-1}\sum_{j=i+1}^{C}\;\left|SSIM\left(COV\left(c_{i}\left(f\left(B\left(X_{R}\right)\right)\right)\right),COV\left(c_{j}\left(f\left(B\left(X_{R}\right)\right)\right)\right)\right)\right|, \end{array}$$
where *B* represents batch-wise data, $c_{i/j}$ generates the categorical output by $c_{i/j}\left(f\left(B\left(X_{R}\right)\right)\right)=f(B\left(X_{R}^{Y_{R}==i/j)}\right)$. COV calculates the covariance matrix of categorical features as in Equation (Equation 3) and |·| takes the absolute value to accelerate the convergence.
(3)$$COV=\frac{1}{N_{B}}\sum_{z=1}^{N_{B}}\left(B_{Z}^{z}-\mu_{Z}\right){\left(B_{Z}^{z}-\mu_{Z}\right)}^{T},\;\;\mathrm{where}\;\mu_{Z}=\frac{1}{N_{B}}\sum_{z=1}^{N_{B}}B_{Z}^{z}$$
where $\mu_{Z}$ is the data mean and $B_{Z}$ is either $c_{i}\left(f\left(B\left(X_{R}\right)\right)\right)$ or $c_{j}\left(f\left(B\left(X_{R}\right)\right)\right)$. The SSIM can be computed in Equation (Equation 4).
(4)$$\begin{array}{c} SSIM\left(B_{1},B_{2}\right)=\frac{\left(2\mu_{B_{1}}\mu_{B_{2}}+C_{1}\right)\left(2\sigma_{B_{1}B_{2}}+C_{2}\right)}{\left(\mu_{B_{1}}^{2}+\mu_{B_{2}}^{2}+C_{1}\right)\left(\sigma_{B_{1}}^{2}+\sigma_{B_{2}}^{2}+C_{2}\right)} \end{array}$$
where $B_{1}=COV\left(c_{i}\left(f\left(B\left(X_{R}\right)\right)\right)\right)$ and $B_{2}=COV\left(c_{j}\left(f\left(B\left(X_{R}\right)\right)\right)\right)$ are batch-wise features, $\mu_{B_{1}},\mu_{B_{2}},\sigma_{B_{1}},\sigma_{B_{2}}$, and $\sigma_{B_{1}B_{2}}$ are mean, standard deviations of domain invariant and specific features batch, and cross-covariance for $\left(B_{1},B_{2}\right)$. $C_{1}$ and $C_{2}$ are two variables to stabilize the division with weak denominator. This loss function is derived from structural similarity index measure (SSIM) [27]. It has the advantages of measuring luminance, contrast, and structural difference between $B_{1}$ and $B_{2}$. Therefore, $L_{S}$ has more capability of measuring the similarity between any two different categorical samples. In addition, the range of the $L_{S}$ is from 0 to 1, where 1 indicates high similarity between batch features and 0 means they are not similar.

During the training, minimizing $L_{S}$ can lead to the minimal similarity between each of the two categories. Hence, it can achieve the inter-class dispersion.

### 3.5. Confident Pseudo Labeling

By combining separation loss with cross-entropy loss, we can improve the discrimination of classifier *f* using training dataset. To improve the performance of the test dataset, we leverage transductive learning to mitigate the difference between the training and test datasets. Transductive learning can train both labeled training data and test samples (without true labels); hence, the difference between them can be minimized [15].

To obtain knowledge from the test dataset, we first generate confident pseudo labels. Previous work either utilized hard pseudo labels or predicted class probability. In contrast to previous approaches, we aim to continuously train the new confident pseudo labeled test data. In this stage, we also take advantage of the initial training classifier *f* to generate initial pseudo labels and examples for the test data. We define a confident pseudo label in the following equation.
(5)$$\begin{array}{cc} C(Y_{T_{P}}^{j})= & \underset{\begin{array}{c} c\in C \end{array}}{arg max}\left\{f_{c}\left(X_{T}^{j}\right)\right\}\;\;\mathrm{if}\max\left(f_{c}\left(X_{T}^{j}\right)\right)>p, \end{array}$$
where C represents confidence. $C(Y_{T_{P}}^{j})$ is the confident label and $C(X_{T}^{j})$ is its corresponding confident sample. Here, $f_{c}\left(X_{T}^{j}\right)$ is the predicted probability in class *c* given the observation $X_{T}^{j}$. max(·) takes the dominant class probability, and it is higher than the threshold *p*, and *p* is between 0 and 1. The confident samples and their confident labels are able to push the decision boundary of classifier *f* toward the test dataset.

We can construct a pseudo label test domain $D_{P}={\left\{X_{P}^{n},Y_{P}^{n}\right\}}_{n=1}^{N_{P}}$, which consists of confident test examples with its confident pseudo labels, where $N_{P}\leq N_{T}$, $X_{P}=C\left(X_{T}\right)$ and $Y_{P}=C\left(Y_{T_{P}}\right)$, and $N_{P}$ is controlled by *p*. $N_{P}=0$ if p=1, and $N_{P}=N_{T}$ if p=0.

However, this pseudo labeling method generates confident pseudo labels with only a single high probability. The classifier *f* can be updated in the early stages of training but may not be able to train more examples on successive iterations since all high probability samples are treated as confident samples. Therefore, we propose to continuously generate confident examples in *T* times adjustment learning so that the classifier *f* could be updated in each adjustment learning. In adjustment learning, the pseudo label test domain becomes: $D_{P}^{t}={\left\{X_{P^{t}}^{n},Y_{P^{t}}^{n}\right\}}_{n=1}^{N_{P^{t}}}$, where $X_{P^{t}}=C\left(X_{T^{t}}\right)$ and $Y_{P^{t}}=C\left(Y_{T_{P}^{t}}\right)$, and *t* is between $\left[1,T\right]$. To remove noisy pseudo labels of the predicted target domain in every *t*, we set the number of *t*-th updated domain $N_{P^{t}}$ is not larger than the target domain sample size $N_{T}$, which means $N_{P^{t}}\leq N_{T}$.
(6)$$\begin{array}{cc} C(Y_{T_{P}^{t}}^{j})= & \underset{\begin{array}{c} c\in C \end{array}}{arg max}\left\{f_{c}\left(X_{T^{t}}^{j}\right)\right\}\;\;\mathrm{if}\max\left(f_{c}\left(X_{T^{t}}^{j}\right)\right)>p_{t}. \end{array}$$ In addition, $C(Y_{T_{P}^{t}}^{j})$ is updated using Equation (Equation 6) with probability threshold $p_{t}$ of every *t*, it also meets the requirements ($p_{t+1}\leq p_{t}$ and $0\leq p_{t}\leq 1$), and we could obtain confident examples and pseudo labels during each *t*-th iteration and the classifier *f* will lean toward the test data. In *T* times iterations, we then form a set of probability threshold as $p_{T}={\left\{p_{t}\right\}}_{t=1}^{T}$. This approach produces confident examples and pseudo labels in each recurrent training interval.

During training, the constructed pseudo labeled test data domain $D_{P}^{t}$ will keep optimizing the trained classifier *f* after minimizing cross-entropy loss and separation loss functions. The pseudo labeled test data are also minimized by the cross-entropy loss. Therefore, the loss function for $D_{P}^{t}$ in each training iteration is given by:(7)$$L_{CE}^{T}=-\frac{1}{N_{P^{t}}}\sum_{n=1}^{N_{P^{t}}}\sum_{c=1}^{C}Y_{P_{c}^{t}}^{n}\log\left(f_{c}\left(X_{P^{t}}^{n}\right)\right),$$
where $N_{P^{t}}$ is the number of confident samples of *t*-th adjustment learning. $Y_{P_{c}^{t}}^{n}\in {\left[0,1\right]}^{C}$ is the confident pseudo labeled binary indicator of each class *c* for the confident sample $X_{P^{t}}^{n}$ in the *t*-th adjustment training, and $f_{c}\left(X_{P^{t}}^{n}\right)$ is also the predicted probability of each class *c* given the input of confident sample $X_{P^{t}}^{n}$.

### 3.6. Categorical Maximum Mean Discrepancy

The proposed confident pseudo labeling process can optimize the network parameters, and it is not necessary to minimize the differences between the training and test data. To reduce the discrepancy between training and test data, we also compute the maximum mean discrepancy (MMD) loss [28], which is a frequently used distance-based loss function that reduces the divergence between the training and test data. However, MMD loss in conventional form focuses on only the marginal distribution alignment, which is more suitable for large domain divergence problems. As shown in Figure 1, the training and test data overlap, suggesting the marginal distribution alignment is not important for these cow teat images. Due to the fuzzy boundaries between different categories, conditional distribution alignment is required. Hence, we propose a categorical MMD (CMMD) loss, which attempts to align the conditional distribution of each category of training and test data.
(8)$$\begin{array}{cc} \; & L_{\mathrm{CMMD}}=\frac{1}{C}\sum_{c=1}^{C}\left(\frac{1}{N_{R^{c}}^{2}}\sum_{i,j}^{N_{R^{c}}}\kappa \left(L_{R^{c}}^{i},L_{R^{c}}^{i}\right)\;+\frac{1}{N_{P^{c}}^{2}}\sum_{i,j}^{N_{P^{c}}}\kappa \left(L_{T^{c}}^{i},L_{T^{c}}^{j}\right)-\frac{2}{N_{R^{c}}\cdot N_{P^{c}}}\sum_{i,j}^{N_{R^{c}},N_{P}^{c}}\kappa \left(L_{R^{c}}^{i},L_{P^{c}}^{j}\right)\right), \end{array}$$
where $N_{R^{c}}$ and $N_{P^{c}}$ are the number of samples in each class of training and confident pseudo labeled test data, $L_{R^{c}}=f\left(X_{R^{c}}\right)$, and $L_{T^{c}}=f\left(X_{P^{c}}\right)$. $X_{R^{c}}$ and $X_{P^{c}}$ are categorical samples. This proposed CMMD loss measures the discrepancy between training and test datasets.

### 3.7. SCTL Model

The framework of our proposed SCTL model is depicted in Figure 2. Combining all loss functions, our model minimizes the following objective function:(9)$$\begin{array}{cc} L(X_{R},Y_{R},X_{T})= & arg min(L_{CE}+\alpha L_{S}+\beta L_{CE}^{T}+\gamma L_{\mathrm{CMMD}}) \end{array}$$
where $L_{CE}$ is the source classification loss, $L_{S}$ is the separation loss, and $L_{\mathrm{CMMD}}$ minimizes the categorical distance between training and test data. $L_{CE}^{T}$ is cross-entropy loss for confident pseudo labeled test data. α, β, and γ are three trade-off parameters. Figure 3 shows a toy example of our SCTL model. The overall training algorithm is shown in Algorithm 1.
**Algorithm 1** Separable Confident Transductive Learning Network. B(·) denotes the mini-batch sets, *I* is the number of iterations. $p_{T}={\left\{p_{t}\right\}}_{t=1}^{T}$, and *T* is the number of adjustment learning. $iter_{t}$ is the *t*-th adjustment learning.1:**Input:** labeled training data $D_{R}={\left\{\left(X_{R}^{i},Y_{R}^{i}\right)\right\}}_{i=1}^{N_{R}}$ and unlabeled test data $D_{T}={\left\{X_{T}^{j}\right\}}_{j=1}^{N_{T}}$2:**Output:** predicted test labels3:**repeat**4:   Derive batch-wise data $\left(B\left(X_{R}\right),B\left(Y_{R}\right)\right)$ and $B(X_{T})$ from $D_{R}$ and $D_{T}$5:   **for** iter=1 **to** *I* **do**6:        Train classifier *f* using Equations (Equation 1) and (Equation 2)7:        **if** $iter=iter_{t}$ **then**8:            Get $p_{t}$9:        **end if**10:      Generate confident pseudo test labels $C(Y_{T_{P}^{t}})$ using Equation (Equation 6)11:      Optimize *f* using Equation (Equation 7)12:      Minimize the differences between training and test data using Equation (Equation 8)13:      Minimize overall loss with Equation (Equation 9)14:   **end for**15:**until** converged16:Make prediction for test samples based on trained classifier *f*

## 4. Experiments

### 4.1. Datasets

We utilize the dataset from [4]. A total of 398 digital images of dairy cows on two commercial New York dairy farms were obtained: farm A milked approximately 1600 Holstein cows in a 60-stall rotary parlor, and farm B milked approximately 4000 Holstein cows in a 100-stall rotary parlor. Thus, our dataset includes 398 dairy cows, and a total of 1529 teat images were extracted in four categories (Score 1, Score 2, Score 3, and Score 4). A total of 380 teat images (around 70 cows) were utilized for the test dataset. For a fair comparison of different algorithms, we split the dataset into training (75%, 1149 images, around 288 cows) and test (25%, 380 images, around 70 cows) datasets. All results are reported based on the test dataset. Table 1 shows the statistics of the teat-end images dataset. Scores 3 and 4 are not common compared with Scores 1 and 2.

### 4.2. Implementation Details

As shown in Figure 2, we utilize seventeen different ImageNet models as the backbone network during the training. The parameters during the training are epochs (100), batch size (16), learning rate (3 × 10^−5^), (α=0.3), (β=1), (γ=0.5), (T=3), and $p_{t}=\left\{0.9,0.8,0.5\right\}$. We report the accuracy of test dataset by: $Accuray=\frac{1}{N_{T}}\sum_{i=1}^{N_{T}}\left(Y_{T_{P}}^{j}==Y_{T}^{j}\right)\times 100.$ We also compare our results with [4] and conduct an ablation study to show the effect of different loss functions on classification accuracy. Since four categories are unbalanced, we also assigned the weight to each class according to the number of each category in the training dataset, and the assigned weights are [0.71,0.65,1.70,15.17] for four categories, respectively. We implement our approach using PyTorch (version 1.7.1, CUDA version: 11.1). The model has trained on a Dell Latitude 7420 laptop (Windows 10) with 16 GB RAM using GeForce 1080 Ti GPU.

### 4.3. Results

As shown in Table 2, we compared the accuracy of seventeen different ImageNet models. We observe that our proposed SCTL with GoogLeNet achieves the highest accuracy when compared with all other models. Moreover, there is consistent improvement across seventeen different ImageNet models, and we achieve 4.9% average improvement. We conclude that our proposed SCTL model improves the performance of ImageNet models. The accuracy of each category of the top-4 highest accuracy model is shown in Table 3. Our model with GoogLeNet has the highest accuracy in the Score 4 category, suggesting our SCTL paradigm is able to handle the unbalanced class problem. Score 4 corresponds to the teat-end condition with the highest degree of hyperkeratosis. This result further indicates the SCTL model can improve inter-class dispersion since SCTL can improve the accuracy in detecting severe teat-end affection even with a small number of training samples. Although the performance of Score 3 is slightly lower than NasNetLarge-SCTL, it is still much higher than DenseNet161-SCTL and DenseNet201-SCTL. The confusion matrixes of these four models are shown in Figure 4. We find that GoogLeNet-SCTL achieves the highest performance, and it has better performance than the other three models in Scores 1 and 4. When compared with earlier work [4], our model improves performance by 15.8%; our SCTL model substantially enhances the accuracy of teat-end image classification datasets. We also notice that the accuracy of “Original” with GoogLeNet, which only minimizes the cross-entropy loss, is still higher than the result from [4]. One possible explanation for the difference is that Porter et al. [4] trained GoogLeNet using MATLAB, while our model uses PyTorch. We also compare results from one transductive learning model (GSM) and three domain adaptation methods (DAN, DCORAl, and CAN). Experimental results show that our GoogLeNet-SCTL still achieves the highest performance. Table 4 and Table 5 display findings from the ablation studies.

To demonstrate the effects of different loss functions ($L_{S}$: “S” (separation loss), $L_{CE}^{T}$: “T” (cross-entropy loss of confident pseudo labeled test data), and $L_{\mathrm{CMMD}}$): “C” (categorical MMD loss) an ablation study in shown in Table 5. Notice that cross-entropy loss is required for the training data. “SCTL-T-S-C” is implemented without $L_{CE}^{T}$, $L_{S}$, and $L_{\mathrm{CMMD}}$ loss. It only reduces training data cross-entropy loss. “SCTL-T-S” minimizes the cross-entropy loss and categorical MMD loss. “SCTL-C” reports results without performing categorical MMD loss. Based on the average accuracy, we find $L_{S}>L_{\mathrm{CMMD}}>L_{CE}^{T}$. Therefore, the proposed separation loss, categorical MMD loss, and confident pseudo labeling approaches are effective in improving the performance of the test dataset. To show the effectiveness of our proposed $L_{S}$, $L_{CE}^{T}$, and $L_{\mathrm{CMMD}}$, we also conducted an ablation study to show different variants of them. In Section 3.4, we utilize the SSIM to measure the similarity between the training and test data, and we take an absolute value to accelerate the convergence. As shown in Table 4, we report the accuracy and the number of convergence of different variants of our proposed separation loss. We find that Jaccard similarity has a lower accuracy than cosine similarity, although it has a longer convergence number. Furthermore, compared with $L_{S}$ without taking the absolute value (w/o abs.), our model achieves high accuracy with the fastest convergence times. When comparing $L_{\mathrm{CMMD}}$ with original MMD loss, our proposed loss function still achieves better accuracy and requires fewer training iterations. Our proposed confident pseudo labeling with adjustment learning is again better than the pseudo label strategy in [20]. Therefore, our proposed loss functions can fast and accurately improve classification accuracy.

### 4.4. Parameter Analysis

There are five hyperparameters α, β, γ, *T*, and $p_{t}$ in our SCTL model. α, β, and γ are three trade-off parameters to balance the weight between separation loss, pseudo labeled test cross-entropy loss, and categorical MMD loss. *T* and $p_{t}$ control the number of adjustment learning and the probability of selecting the confident examples, respectively. To obtain the optimal parameters, we use GoogLeNet as the backbone network. We first show the influence of α, β, and γ on test data accuracy. α, β, and γ are selected from {0.1,0.2,0.3,0.4,0.5,0.6,0.7,0.8,0.9,1} and fix one parameter while varying the others. As shown in Figure 5a, the x-axis represents that different values of α, β, and γ. We observed that the test data accuracy achieves the highest value when α=0.5, β=1, and γ=0.3, respectively. *T* is selected from {1,2,3,4,5,6,7,8,9,10}, and $p_{t}$ is selected from {0.9,0.8,0.7,0.6,0.5}. Since we need to obtain confident examples, $p_{1}$ should be a very high probability. Thus, we set $p_{1}\geq 0.5$. For t>2, $p_{t}$ is selected from {0.9,0.8,0.7,0.6,0.5} and require $p_{t}\geq p_{t+1}$. As shown in Figure 5b, we observed that our model achieves the highest test data accuracy when T=3. We then examined how different $p_{t}$ values affect the accuracy in Figure 6. We observed that when $p_{t}=\left\{0.9,0.8,0.5\right\}$, the highest accuracy in the test data is achieved. By carefully examining these parameters and their influence on overall performance, we find the best hyperparameters for our SCTL model are: α=0.5, β=1, γ=0.3, T=3, and $p_{t}=\left\{0.9,0.8,0.5\right\}$.

### 4.5. Feature Visualization

To further demonstrate the effectiveness of different loss functions, we utilize t-SNE [9] to visualize the deep features of network activations in 2D space. As shown in Figure 7a, we cannot observe four distinctive classes if we only minimize the cross-entropy loss. From Figure 7b to Figure 7g, the four classes become more distinctive after adding separation loss, pseudo labeled test cross-entropy loss, and categorical MMD loss. Comparing Figure 7b,c with Figure 7d, the four categories cannot be correctly classified if we only train the network with a single loss (especially Score 4 in the test data which are missing). There is also contamination between classes 1 and 2 among these three figures. These two issues are ameliorated if we train the model with two losses (from Figure 7e to Figure 7g). Figure 7g has a similar trend as Figure 7h with less class divergence. Finally, with SCTL (Figure 7h), we see inter-class dispersion and intra-class compactness of the test dataset.

## 5. Discussion

### 5.1. Relationship between ImageNet Accuracy and Teat-End Accuracy

Previous work [40] noted that ResNet and DenseNet are usually the better neural networks for transfer learning, and a better ImageNet model can produce better features for domain adaptation [41], which is one special case of transductive learning. We explore how different ImageNet models affect the teat-end classification accuracy, their correlation score, and the $R^{2}$ value as per [41]. As shown in Figure 8, both correlation score and $R^{2}$ value are low, which suggests no strong relationship between ImageNet model accuracies and teat-end classification accuracies. This result differs from [40,41], suggesting the optimal ImageNet model for teat-end classification may not be based on the ImageNet model with the highest accuracy.

### 5.2. What Can We Draw from Our Experiments?

Fine-tuning different ImageNet models for transfer learning has been one of the most popular methods for image classification problems. However, choosing the optimal ImageNet for a given dataset remains a challenge. For our dataset, the teat images vary from the ImageNet datasets and thus there is no strong relationship between ImageNet model accuracy and teat-end classification accuracy. As shown in Table 2, GoogLeNet unexpectedly achieved the highest performance. This suggests there is value in using ImageNet models with lower memory size when first exploring such techniques, such as (SqueezeNet and GoogLeNet) for transfer learning if the image data are very different from the pre-trained ImageNet dataset. If image data are very similar to the ImageNet images, there may be value in using more accurate networks such as Xception and EfficientNet.

### 5.3. Advantages and Limitations

There are several advantages of our proposed SCTL model. First, our proposed separation loss enlarges the difference between different categories and leads to greater inter-class dispersion. Second, we generate high confident pseudo labels for test data in three times adjustment learning to optimize the network with pseudo labels information. Last, we propose a categorical MMD loss to reduce the divergence between training and test data. By aggregating all three of these novel loss functions, our SCTL model can enhance the performance of the teat-end image classification problem.

One limitation of our work is that we have a small sample size of teat-end images (1529 images). Especially, the category Score 4 is unbalanced. However, Score 4 corresponds to the severe hyperkeratosis of the teat-end, which is less prevalent in the study population when compared with the other three categories. As for future work, aside from collecting more data, improving the pseudo label quality of the test dataset can be a useful technique to further improve performance. Our SCTL model can be applied to other image classification tasks (e.g., teat skin condition assessments). However, five hyperparameters, α, β, γ, *T*, and $p_{t}$, should be adjusted according to different datasets.

## 6. Conclusions

In this paper, we propose a separation confident transductive learning model for teat-end image classification. We first propose a separation loss to enlarge the differences between different categories. We then generate confident labels for the test data using adjustment learning to optimize the network. Finally, we employ transductive learning to minimize the divergence between the training and test data with a categorical MMD loss. Although the level of affection of cows’ teats can influence the performance of our SCTL model, we demonstrate that the proposed SCTL model can achieve higher accuracy when compared with ImageNet transfer learning models. We believe that through the aid of SCTL, the detection of hyperkeratosis is feasible in the commercial dairy farm setting. Our approach offers the opportunity for more frequent and automated teat-end condition assessments. Such an automated hyperkeratosis detection method may help farmers mitigate the risks of intramammary infections, decrease the use of antimicrobials, control the costs associated with detecting and managing mastitis, and improve the quality of life of dairy cows and farmers.

## Figures and Tables

**Figure 1 animals-12-00886-f001:**
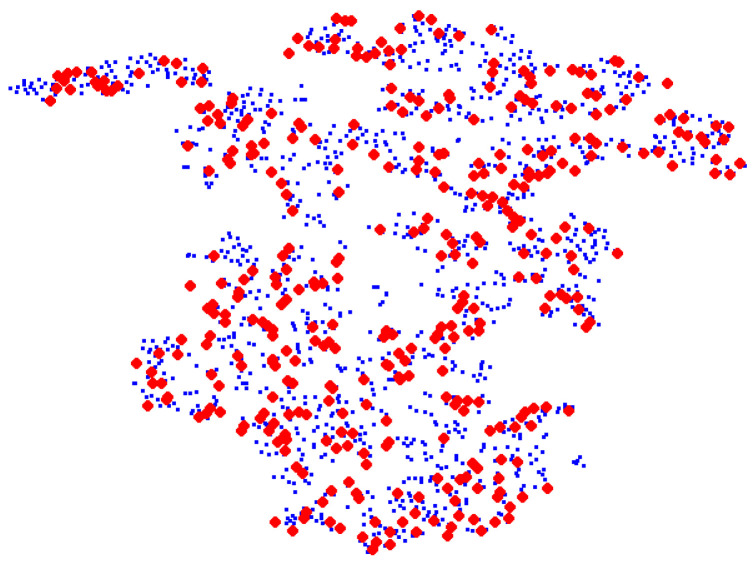
t-SNE [9] view of training (blue) and test (red) dataset from a retrained GoogLeNet [4]. Different categories are mixed together after training.

**Figure 2 animals-12-00886-f002:**
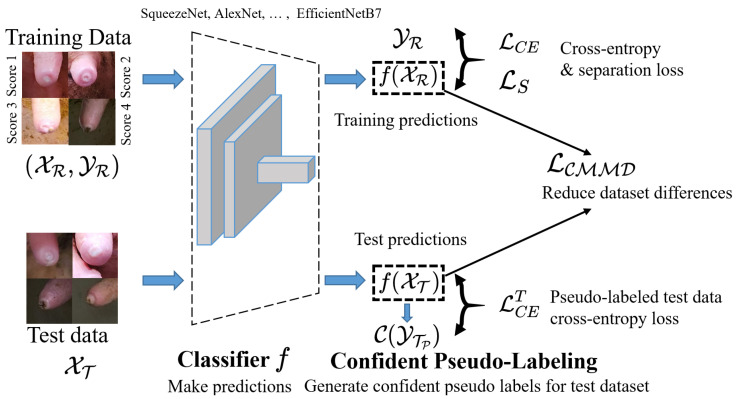
The learning scheme of our proposed SCTL model. We first fine-tune the classifier *f* from seventeen well-known ImageNet models and then make predictions for the training ($X_{R}$) and test datasets ($X_{T}$). For the training data, we minimize the typical cross-entropy loss ($L_{CE}$) and the separation loss ($L_{S}$) to improve the inter-class dispersion. For the test data, we generate confident pseudo labeled examples ($\left\{C\left(X_{T^{t}}\right),C\left(Y_{T_{P}^{t}}\right)\right\}$) in the *t* adjustment learning, and then we minimize the pseudo labeled test data cross-entropy loss ($L_{CE}^{T}$). To reduce the dataset differences, we also develop a categorical maximum mean discrepancy loss $L_{\mathrm{CMMD}}$ to improve intra-class compactness.

**Figure 3 animals-12-00886-f003:**
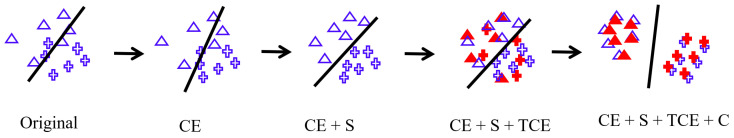
A toy example of our SCTL learning paradigm. The blue color is the training data, and the red color is the test data. “Original” is the binary classification problem, “CE” refers to performing cross-entropy loss in training data, “CE + S” minimizes the proposed separation to enlarge differences between the two classes. “CE + S + TCE” can additionally minimize the pseudo labeled test data using the cross-entropy loss. “CE + S + TCE + C” adds another categorical maximum mean discrepancy loss to reduce the divergence between training and test data and form our SCTL model.

**Figure 4 animals-12-00886-f004:**
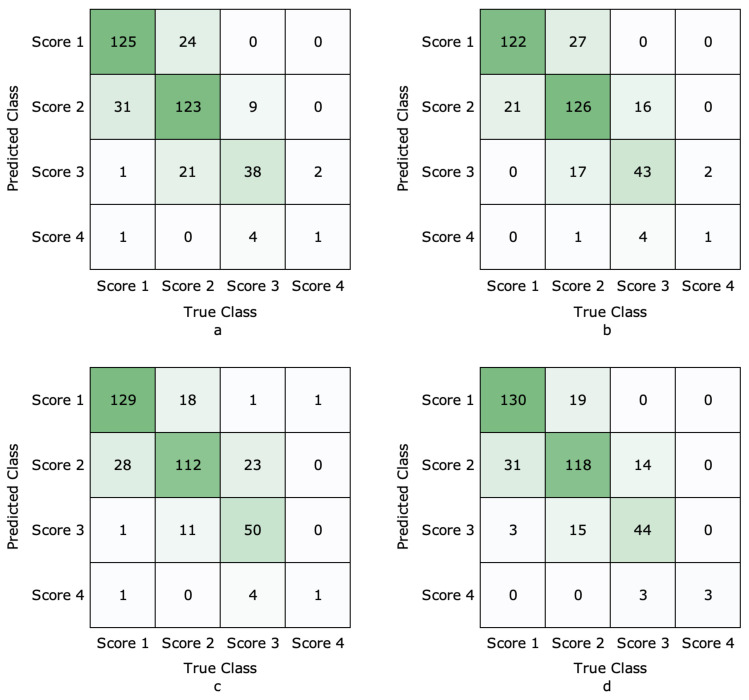
Confusion matrix of DenseNet161-SCTL, DenseNet201-SCTL, NasNetLarge-SCTL, and GoogLeNet-SCTL models. (**a**) DenseNet161-SCTL, (**b**) DenseNet201-SCTL, (**c**) NasNetLarge-SCTL, (**d**) GoogLeNet-SCTL.

**Figure 5 animals-12-00886-f005:**
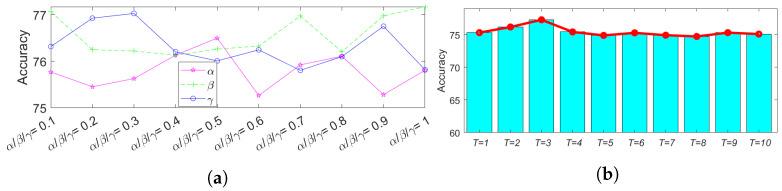
Parameter analysis for α,β, γ, and *T*. The effect of different α,β, and γ on test accuracy is shown in (**a**), and effect of different *T* on test accuracy is shown in (**b**). When α=0.5, β=1, and γ=0.3, and T=3, accuracy is highest.

**Figure 6 animals-12-00886-f006:**
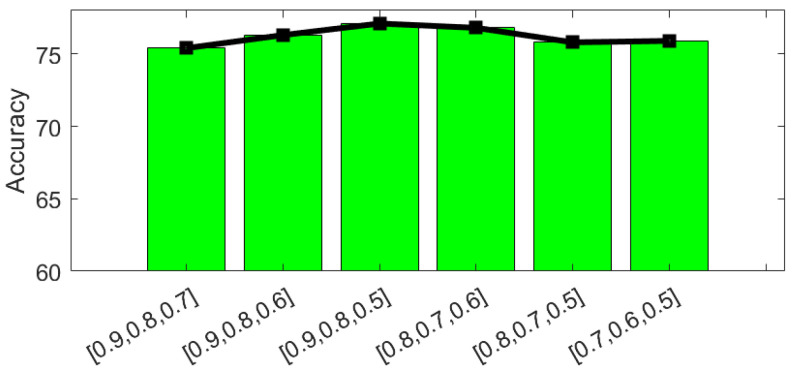
Effect of different $p_{t}$ on test accuracy.

**Figure 7 animals-12-00886-f007:**
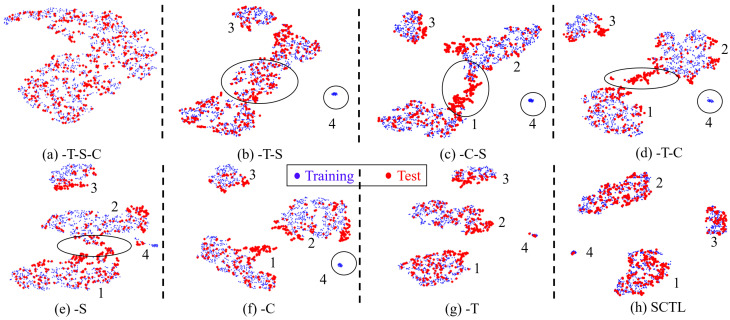
t-SNE view of training and test dataset with different loss functions. Blue color represents training dataset and red represents test dataset. Numbers 1 to 4 mean the location of four classes (Score 1 to Score 4).

**Figure 8 animals-12-00886-f008:**
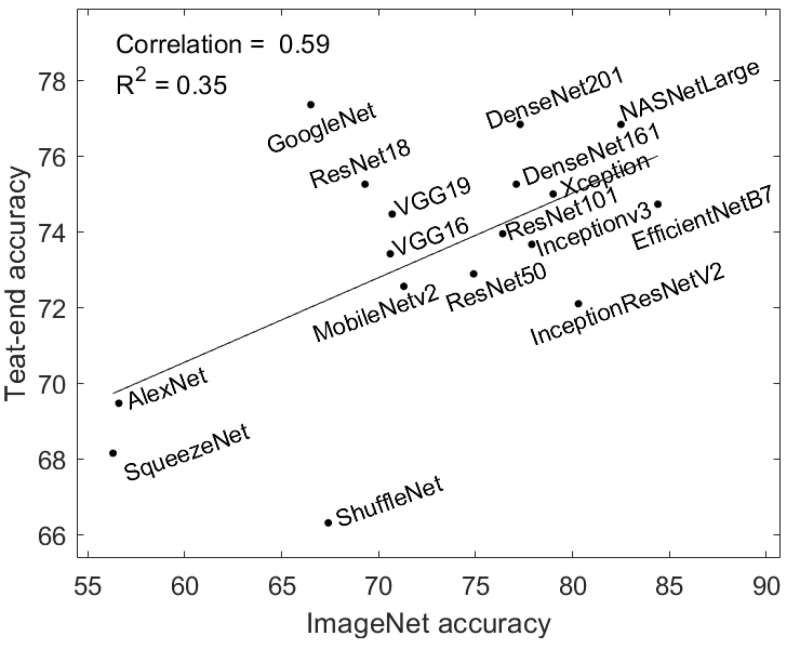
ImageNet models’ accuracy and teat-end classification accuracy. There is no strong relationship between them since both correlation and $R^{2}$ have a low value.

**Table 1 animals-12-00886-t001:** Statistics of training and test data.

Label	Training		Test	
	Number	Percentage	Number	Percentage
Score 1	450	39.2	149	39.2
Score 2	491	42.7	163	42.9
Score 3	187	16.3	62	16.3
Score 4	21	1.8	6	1.6

**Table 2 animals-12-00886-t002:** Accuracy of different ImageNet models and the ablation study. GoogLeNet demonstrated the highest accuracy and greatest improvement when adopting SCTL.

Networks	SCTL	Original	Improvement
SqueezeNet [29]	68.2	67.4	0.8
AlexNet [5]	69.5	67.1	2.5
**GoogLeNet [6]**	**77.6**	64.2	13.4
ShuffleNet [30]	66.3	64.5	1.8
ResNet18 [31]	75.3	**72.6**	2.7
VGG16 [32]	73.4	66.8	6.6
VGG19 [32]	74.5	67.9	6.6
MobileNetv2 [33]	72.6	70.5	2.1
ResNet50 [31]	72.9	70.8	2.1
ResNet101 [31]	74.0	68.4	5.6
DenseNet161 [34]	75.5	72.9	2.6
DenseNet201 [34]	76.8	70.5	6.3
InceptionV3 [7]	73.7	71.6	2.1
Xception [35]	75.0	70.5	4.5
InceptionResNetV2 [36]	72.1	66.3	5.8
NasNetLarge [8]	76.8	68.2	8.6
EfficientNetB7 [8]	74.7	71.3	3.4
Average	**73.5**	**68.6**	4.9
GoogLeNet [4]	-	61.8	**15.8**
GSM [15]	-	65.2	12.4
DAN [37]	-	62.1	15.5
DCORAL [38]	-	63.8	13.8
CAN [39]	-	67.4	10.2

**Table 3 animals-12-00886-t003:** Accuracy of each class in the test dataset. Shown in bold are the highest performance of a network for each score. Note that no single network achieves the highest accuracy for all scores.

Networks	Score 1	Score 2	Score 3	Score 4	Ave.
DenseNet161-SCTL	83.9	75.5	61.3	16.7	75.5
DenseNet201-SCTL	85.2	**77.3**	62.9	0.0	76.8
NasNetLarge-SCTL	86.6	71.2	**74.2**	16.7	76.8
**GoogLeNet-SCTL**	**87.3**	72.4	71.0	**50.0**	**77.6**

**Table 4 animals-12-00886-t004:** Ablation study of $L_{S}$ and $L_{\mathrm{CMMD}}$.

Method	Accuracy	# Convergence
Cosine similarity	75.3	92
Jaccard similarity	74.5	83
w/o abs.	76.8	78
MMD	76.3	75
PL [20]	75.5	83
**SCTL**	77.6	70

**Table 5 animals-12-00886-t005:** Ablation study of different loss functions on test accuracy.

Loss	Accuracy
SCTL-T-S-C	64.2
SCTL-T-S	66.6
SCTL-T-C	71.1
SCTL-S-C	72.9
SCTL-T	74.7
SCTL-S	72.1
SCTL-C	74.4
**SCTL**	77.6

## Data Availability

We evaluate our model using a dataset from previous work [4]. Source code, accessed 25 March 2022, is available at https://github.com/YoushanZhang/SCTL.

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
