# Peer review of "Separable Confident Transductive Learning for Dairy Cows Teat-End Condition Classification"

_animals, 2022, doi:10.3390/ani12070886_

Round 1
Reviewer 1 Report
Very well written article, with important scientific contributions to milk business. Congratulations to all team researchers!
Author Response
Thanks for your time and positive feedback from Reviewer 1! We also addressed comments from other reviewers.
Reviewer 2 Report
It would be worth writing in the article what was the cognitive (scientific) goal of the research and what utilitarian (useful) goals were set by the authors? In my opinion, it would be worth summarizing the review of the state of knowledge in the Introduction by formulating the research problem (The research problem is ...). The research problem can be associated with an indication of a gap in the current state of knowledge.
Why is the last formula on page 5 not marked. In this case, the marking (5) falls. A similar formula is given on page 6, but assigned the reference number (5). Therefore, I believe that the formula on page 5 must be marked in the order of the presented mathematical formulas.
I would like to ask what is / was the measurement error in the research verifying the theoretical considerations? Was such an error determined?
If data were collected on dairy farms, which was referred to in the acknowledgments to farmers, then what was the characteristics of dairy cow herds on farms? From how many cows has the data been collected for model verification and testing?
Much of the paragraph in the Conclusions chapter is, in my opinion, not conclusions, but merely statements of what the authors did. Only the final sentences in the Conclusions chapter mention the achievement of greater accuracy when comparing the models.
It would be useful to write in the article what practical benefits result from the automatic classification of teat-end conditions for dairy cows. Are these benefits for farmers or are the results of the study only broadening the cognitive (theoretical) knowledge? What is the practical application of the research?
Reviewer 3 Report
Manuscript ID: animals-1641304
Manuscript title: Separable Confident Transductive Learning for Dairy Cows
Teat-End Condition Classification
I hope the authors find this comments useful.
General comments
The main objective of this experiment was to propose a new methodology to classify alterations of the teat end of dairy cows based on digital images. The proposal is correct since it seeks to improve the results obtained by other digital methodologies and would serve to simplify the control of the health status of the teats, especially in farms with a high number of animals.
Specific comments
Lines 55-71:
The correspondence of the 4 classes of hyperkeratosis of the teat used with the "standard" classification that is usually used in the sector should be clarified, beyond sending the reader to a reference.
Table 3 and Discussion:
It can be seen that the goodness of the results of the new proposal clearly improves in the case of Score 4, which corresponds to the teats with the highest degree of affection. These results should be discussed in depth since they could guide future new proposals for improvement.
Conclusion:
The goodness of the results obtained depending on the level of affection of the teats should be mentioned in this section.

Author Response
Please see the attachment.
There is an attached pdf file, which is named ``peer-review-18202076.v1.pdf". It seems that it is a review for another paper. Please let us know if it is a wrong attachment. Thanks!

Reviewer 4 Report
The presented manuscript poses a new method based on a neural-network transfer learning strategy, referred as 'Separable Confident Transductive Learning' aiming to its use in the classification of dairy cows teat-end condition clasification from digital images of these appendices.
The document is written in a very good English. It is fairly well organized, and it includes an Abstract, a first section containing an Introduction, followed by a Related Work, Methods, Experiments, Discussion and Conclusion sections, respectively, finishing with the listing of References used.
As it is, in my opinion the manuscript shows enough value, and it seems to describe an interesting proposal for the intended classification of teat-end images.
Aiming to improve the quality of the work and without any intent to underrate neither its accuracy nor its contributions, I would like to make the following suggestions:
Lines 16-18: The sentence 'Machine milking can affect teat canal integrity and lead to increased teat-end callosity- this can increase the risk of bacterial infections.' is difficult to understand because of the second dash sign.
Line 28: The sentence start could be changed to 'The indistinct boundaries of the classes lead to'.
Lines 30-47 might be moved to the Methods section, as the Introduction should be used to contextualize the work and to justify the need to addess it.
Lines 48-52 (plus Figures 1 and 2) might belong to the Discussion section.
Line 63: Perhaps 'are a time-consuming' could be written as 'are time-consuming'.
Lines 70-71: Maybe the sentence 'The overall accuracy of our original approach on test data showed promise but also highlighted the need for improvement' is not clear enough and should be revised.
Line 82: Perhaps 'and test are not' could be written as 'and test sets are not'.
Figure 3: This is too simplistic and poorly informative, needing of a long description in its label to help to understand it. It is recommended to redraw it in a more detailed way, showing the flows of data and its structures as it moves within the system.
Lines 126-127: The sentence 'As shown in Fig. 1, we can find different classes of training and test which mixed together.' has a not very clear meaning and perhaps must be rephrased.
Equation 3: An space seems to be missing between the word 'where' and the 'mu' greek letter. Two lines below, there is a typo in 'compuated'.
Paragraph starting in line 155: The sentence 'Therefore, we propose to continuously generate confident examples in T times adjustment learning.' is not clear enough.
Line 165: Perhaps the sentence 'are number of sample in each class' could be written as 'are the number of samples in each class'.
Algorithm 1: The description is not informative enough, and too much loaded with defining symbols. it is not mentioned that pseudocode is used. A graphic diagram could be helpful instead to better communicate it.
Lines 174, 176 and others: Amounts are better written as 1,529 , 1,149 , ... using commas as separators.
Lines 177, 190, 195 and others: Abbreviating 'Table' as 'Tab.' is not very useful or convenient, suggesting keeping the full word as a better choice.
Line 181: What were the criteria used to select the seventeen different ImageNet models used? Why 17? Please, explain.
Line 188: The description of the software and hardware components is to be included to a detailed enough level, so that the experiments could be replicated and benchmarked if necessary.
Line 190: Perhaps 'we compare accuracy' could be better written as 'we compared the accuracy'.
Line 210: Table 5 clearly overlaps this text line.
Line 254: The references to 'Fig. 72(b)' and 'Fig.72(g)' perhaps should be written as 'Fig. 7(b)' and Fig. 7(g)'.
Lines 274-275: The sentence 'Fine-tuning different ImageNet models for transfer learning been one of the most popular methods for image classification problems.' needs to be rewritten for a better understanding.
Perhaps the 'Conclusion' section could be renamed to 'Conclusions'. It seems to be quite short, and it no doubt would gain weight if extended to highlight the main achievements with some more detail. Additionally, some mentions to future lines of work could be very useful to contextualize the work within its field of study.
There seems to be a lack of attention across the whole document to the limitations and caveats of the approach being used, which might bias the reader towards a more favourable interpretation of the achievements, instead of modulating according to the scope information provided.
Line 319: The 'References' label is missing.
Round 2
Reviewer 3 Report
The authors have satisfactorily addressed the proposals made by the reviewer
Author Response
Thanks for your time and feedback from Reviewer 3!
Reviewer 4 Report
The Authors have incorporated most of the suggestions made, and addressed the majority of the issues pointed to in the previous revision.
However, I think there are still two matters that would require further work:
1. Figure 2 (former Figure 3) is still poorly informative. An effort was made on the topic by extending the descriptive label, but still the graphical part is too simplistic, in my opinion, and I would still recommend to re-elaborate it.
2. The information on the software packages (operating system, software versions, ...) used was not detailed in the text as suggested in my former comments.
Author Response
Thanks for the feedback from Reviewer 4! We highlight all changes with magenta color in the revised manuscript.
Point 1: Figure 2 (former Figure 3) is still poorly informative. An effort was made on the topic by extending the descriptive label, but still the graphical part is too simplistic, in my opinion, and I would still recommend to re-elaborate it.
Response 1: We re-draw Figure 2 with more arrows to show the details of our model.
Point 2: The information on the software packages (operating system, software versions, ...) used was not detailed in the text as suggested in my former comments.
Response 2: We added more details of the experiments about the system and software versions (Line 228-229).
We implement our approach using PyTorch (version 1.7.1, CUDA version: 11.1). The model has trained on a Dell Latitude 7420 laptop (Windows 10) with 16 GB RAM using GeForce 1080 Ti GPU.